# Fragmented QRS complex in patients with systemic lupus erythematosus at the time of diagnosis and its relationship with disease activity

**Masahiro Hosonuma**[1], **Nobuyuki Yajima**[1,2,3]*, **Ryo Takahashi**[1], **Ryo Yanai**[1], **Taka-aki Matsuyama**[4], **Eiji Toyosaki**[5], **Jumpei Saito**[6], **Kengo Kusano**[7], **Hiroshi Morita**[8]

1 Division of Rheumatology, Department of Medicine, Showa University School of Medicine, Tokyo, Japan, 2 Department of Healthcare Epidemiology, Kyoto University Graduate School of Medicine and Public Health, Kyoto, Japan, 3 Center for Innovative Research for Communities and Clinical Excellence, Fukushima Medical University, Fukushima, Japan, 4 Department of Legal Medicine, Showa University, School of Medicine, Tokyo, Japan, 5 Division of Cardiology, Department of Medicine, Showa University School of Medicine, Tokyo, Japan, 6 Division of Cardiology, Showa University Northern Yokohama Hospital, Yokohama, Japan, 7 Division of Arrhythmia and Electrophysiology, Department of Cardiovascular Medicine, National Cerebral and Cardiovascular Center, Osaka, Japan, 8 Department of Cardiovascular Therapeutics, Okayama University Graduate School of Medicine, Dentistry and Pharmaceutical Sciences, Okayama, Japan

* n.yajima@med.showa-u.ac.jp

**Data Availability Statement:** All relevant data are within the manuscript and its Supporting Information files.

## Abstract

### Objective

Cardiovascular disease is an important contributor to the mortality rate of patients with systemic lupus erythematosus (SLE), which is related to SLE disease activity. Fragmented QRS (fQRS) complexes, defined by additional spikes in the QRS complex, are useful for identifying myocardial scars on electrocardiography and can be an independent predictor of cardiac events. We aimed to assess the relationship between disease activity in patients with SLE and fQRS at the time of diagnosis.

### Methods

Forty-four patients with SLE were included. Patients with cardiac diseases, other rheumatic diseases, and prior treatment at the time of electrocardiography measurement were excluded. The appearance of fQRS represented exposure. The primary outcome was SLE Disease Activity Index 2000 (SLEDAI-2K). Multiple regression analysis was conducted to assess the association between fQRS and SLEDAI-2K adjusted for age, sex, and time from the estimated onset date to the date of diagnosis.

### Results

Among patients with SLE at diagnosis, 26 (59.1%) had fQRS. The median SLEDAI-2K was 18 (interquartile range [IQR], 12–22) and 9 (IQR, 8–15) in the fQRS(+) and fQRS(-) groups, respectively. SLEDAI-2K was significantly higher in the fQRS(+) group than in the fQRS(-) group (regression coefficient, 2.69; 95% confidence interval, 0.76–4.61; p = 0.008).

**Funding:** The author(s) received no specific funding for this work.

**Competing interests:** Hiroshi Morita is affiliated with the endowment department of Japan Medtronic, Inc. The other authors declare that there is no conflict of interest. This does not alter our adherence to PLOS ONE policies on sharing data and materials.

## Conclusion

Our results suggested that fQRS(+) patients with SLE had high disease activity. fQRS could likely detect subclinical myocardial involvement in patients with SLE and predict long-term occurrence of cardiac events.

## Introduction

Systemic lupus erythematosus (SLE) is an inflammatory autoimmune disease of unknown etiology that can affect any organ. In patients with SLE, cardiovascular diseases (CVDs) including pericarditis, myocarditis, coronary artery disease (CAD), and endocarditis are major causes of morbidity and mortality [1,2]. Patients with SLE are at a significantly higher risk for CVD than the general population; furthermore, SLE is an independent predictor of heart failure [3]. Traditional risk factors cannot sufficiently explain CVD in this patient population [4]. Some studies have reported that SLE disease activity is associated with the occurrence of myocarditis and CAD [5,6]. Recently, asymptomatic myocarditis and CAD were identified using cardiovascular magnetic resonance (CMR) imaging [7]. It was also reported that myocardial edema, defined by an increased T2 ratio on CMR as myocardial infarction and inflammation, is significantly more evident in patients with SLE who have high disease activity than in other groups [8,9]. However, CMR is not readily available and is expensive. In addition, very few clinicians possess the required expertise to perform this modality. Depending on the patient's condition, the use of CMR can be restricted. Cardiac involvement in the absence of typical cardiac symptoms detected via CMR can be missed on transthoracic echocardiography (TTE); therefore, a new routine indicator is required [10].

Resting electrocardiogram (ECG) is inexpensive and non-invasive and can be performed routinely by a rheumatologist. There are no limitations to its use regarding patient condition, and it is not operator-dependent. Fragmented QRS (fQRS) is a convenient marker of myocardial scarring on ECG and is defined by additional spikes within the QRS complex [11]. The fQRS may be caused by zigzag conductions around the myocardium previously scarred by ischemia or inflammation [12]. It is useful for identifying myocardial scars such as those resulting from CAD and cardiac sarcoidosis, for identifying high-risk patients with various cardiac diseases, and for predicting sudden cardiac death in the general population [11,13–17]. The prevalence of fQRS appears to be higher in patients with rheumatic diseases, such as rheumatoid arthritis (RA), systemic sclerosis (SSc), ankylosing spondylitis, and Behçet's disease, than in controls, and similar findings have been reported in patients with SLE [18–22].

To the best of our knowledge, the appearance of fQRS with untreated SLE at the time of diagnosis and the relationship between disease activity and fQRS have not been reported previously. We hypothesized that fQRS would be expressed more frequently in patients with SLE and high disease activity, thereby representing subclinical myocardial involvement. This study aimed to assess the relationship between disease activity in patients with SLE and fQRS at the time of diagnosis.

## Materials and methods

### Patient selection

In the retrospective review of the medical records in the Showa University Hospital and Showa University Koto Toyosu Hospital from January 2010 to December 2017, we identified patients

who were aged >15 years, diagnosed with SLE, and underwent ECG at the time of diagnosis. Participants who satisfied at least 4 of the 11 American College of Rheumatology (ACR) criteria from 1997 were included. Patients who underwent treatment prior to ECG measurement and those with ischemic heart disease, severe valvular disease, congenital heart disease, cardiomyopathy, history of arrhythmia, hepatic failure, RA, SSc, and abnormal serum electrolytes were excluded. We used the process of sequential sampling as our sampling method. The study was conducted in accordance with the principles outlined in the Declaration of Helsinki and was approved by the ethical review committee of Showa University School of Medicine (approval numbers 2556). All patient information was anonymized and de-identified prior to analysis.

## Data collection

The patients' demographic data including sex, age at the time of diagnosis, blood pressure, smoking status, and comorbid conditions such as hypertension, treatment for diabetes mellitus, and dyslipidemia were collected at the time of diagnosis. Patients were considered to have comorbid hypertension if they were using antihypertensive drugs such as diuretics, beta-blockers, calcium channel blockers, angiotensin-converting enzyme inhibitors, and angiotensin type II receptor blockers. Blood test results and urinalysis data at the time of the SLE diagnosis obtained at the time closest to the ECG measurement and before treatment intervention were included in the analysis. C-reactive protein, low-density lipoprotein cholesterol, high-density lipoprotein cholesterol, triglycerides, uric acid, glycated hemoglobin (HbA1c, measured according to the National Glycohemoglobin Standardization Program), complement (hemolytic complement activity, complement 3, complement 4), C1q-binding immune complexes (IC-C1q), anti-dsDNA antibody, anti-β2-glycoprotein I antibody, anti-SS-A/Ro antibody, and anti-U1-RNP antibody levels were investigated, and urinalysis (proteinuria, casts, hematuria, and pyuria analysis) was performed. The Framingham Risk Score used to estimate the 10-year risk for developing coronary heart disease was evaluated [23]. SLE Disease Activity Index 2000 (SLE-DAI-2K) at the time of the ECG measurement was evaluated by the attending rheumatologist. Morbidities with end-organ involvement including cutaneous manifestations, arthritis, myositis, pericarditis, pleuritis, lupus enteritis, lupus cystitis, vasculitis, renal disorders, neurologic disorders, and hematologic disorders (leukopenia, thrombocytopenia) were documented. Renal disorders were defined as any of the following: renal biopsy indicating lupus nephritis, nephrotic syndrome, increase in serum creatinine level >1.5-times the baseline, proteinuria, urinary casts, hematuria, or pyuria in the absence of other causes. Lupus enteritis was defined according to colonoscopy findings, computed tomography findings, or patient reports of abdominal somatic pain or hematochezia in the absence of other causes. Lupus cystitis was defined by patient reports of bowel or urinary symptoms, or as hydronephrosis in the absence of other causes. Other disease manifestations were defined according to SLEDAI-2K definitions. The estimated date of onset was defined as the day when one or more ACR classification criteria items were reported. Results of TTE examinations performed at the time of the SLE diagnosis were evaluated. Left ventricular (LV) end-diastolic and end-systolic dimensions were measured in the parasternal long-axis view with the M-mode cursor positioned appropriately. The LV ejection fraction (EF) was measured in accordance with the Simpson's method. The right ventricle systolic pressure (RVSP) was calculated using the tricuspid regurgitant velocity.

## Exposure

Exposure was defined as the appearance of fQRS at the time of diagnosis. Results of a resting baseline 12-lead ECG (low-pass filter, 150 Hz; paper speed, 25 mm/s; voltage, 10 mm/mV;

Model ECG 2550; Nihon Kohden, Tokyo, Japan) were recorded before initiating drug therapy. In fQRS (+) patients, the appearance of fQRS was evaluated again on electrocardiogram after immunosuppressive therapy. All ECGs were evaluated at 400% magnification by 2 experienced cardiologists who were blinded to patient characteristics and outcomes. Differences in ECG readings were discussed until an agreement was reached. The fQRS complex was defined by the presence of an additional R wave (R') or notching in the nadir of the R wave or the S wave or >1 R' (fragmentation) in 2 contiguous leads corresponding to the territory of a major coronary artery during a normal QRS interval (Fig 1, S1 Fig). Complete bundle branch block (BBB) patterns (QRS ≥120 ms) and incomplete right BBBs were excluded.[11]

### Outcome measures

The primary outcome was disease activity. SLEDAI-2K, a common method of evaluating disease activity, was used at the time of ECG measurement by an attending rheumatologist who was blinded to the ECG findings. Secondary outcomes included complement and anti-dsDNA antibody levels and end-organ involvement.

### Statistical analysis

Categorical data are described as numbers with proportions (%) and were compared using Fisher's exact test. Continuous data are expressed as means with standard deviations (SD) or as medians with interquartile ranges (IQR), as appropriate, and were compared using the Wilcoxon signed-rank test. Inter-observer variabilities were assessed using Cohen's kappa coefficient. During the main analysis, a multiple regression analysis was conducted to assess the association between fQRS and SLE activity after adjusting for age, sex, and period from the estimated date of onset to the date of diagnosis. During secondary analysis, we performed a multilinear regression analysis to examine the correlations between fQRS and serological markers related to SLE activity (complement and anti-dsDNA antibody levels) and a logistic regression analysis to examine the correlations between fQRS and end-organ involvement under the same conditions as described previously. We did not perform multivariate analyses of end-organ involvements that occurred less frequently. Three sensitivity analyses were performed. First, we analyzed all ECG findings interpreted by the 2 cardiovascular physicians (ET and JS) as exposure. Furthermore, inter-observer variabilities between the cardiologists were assessed. Next, the ECG results were evaluated and analyzed as the main outcome at 100% magnification on paper. Additionally, inter-observer variabilities were assessed at 400% and 100% magnifications and compared. Finally, we excluded patients with hypertension, treated diabetes mellitus, and treated dyslipidemia because of their cardiovascular risks and analyzed the main outcome as the result. Missing data were not imputed. All statistical tests were 2-sided, and significance was defined as p<0.05. Analyses were performed using JMP® Pro, version 14.0.0. (SAS Institute Inc., Cary, NC, USA).

### Results

A total of 44 participants were enrolled (Fig 2). The mean age was 39.5 years, and 37 (84.1%) of the participants were women. The median SLEDAI-2K was 13.5 (IQR, 10–20), and the median period from the estimated date of onset to the date of diagnosis was 3 months (IQR, 2–15). Twenty-six patients (59.1%) had fQRS at the time of diagnosis, 18 patients were followed, and 6 patients (33.3%) disappeared fQRS after immunosuppressive therapy. The mean follow-up period was 27.5 months (IQR, 10.5–42.5). The clinical and demographic characteristics of fQRS(+) and fQRS(-) patients are summarized in Table 1. The SLEDAI-2K results and the number of men were significantly higher in the fQRS(+) group (p<0.001 and p = 0.031,

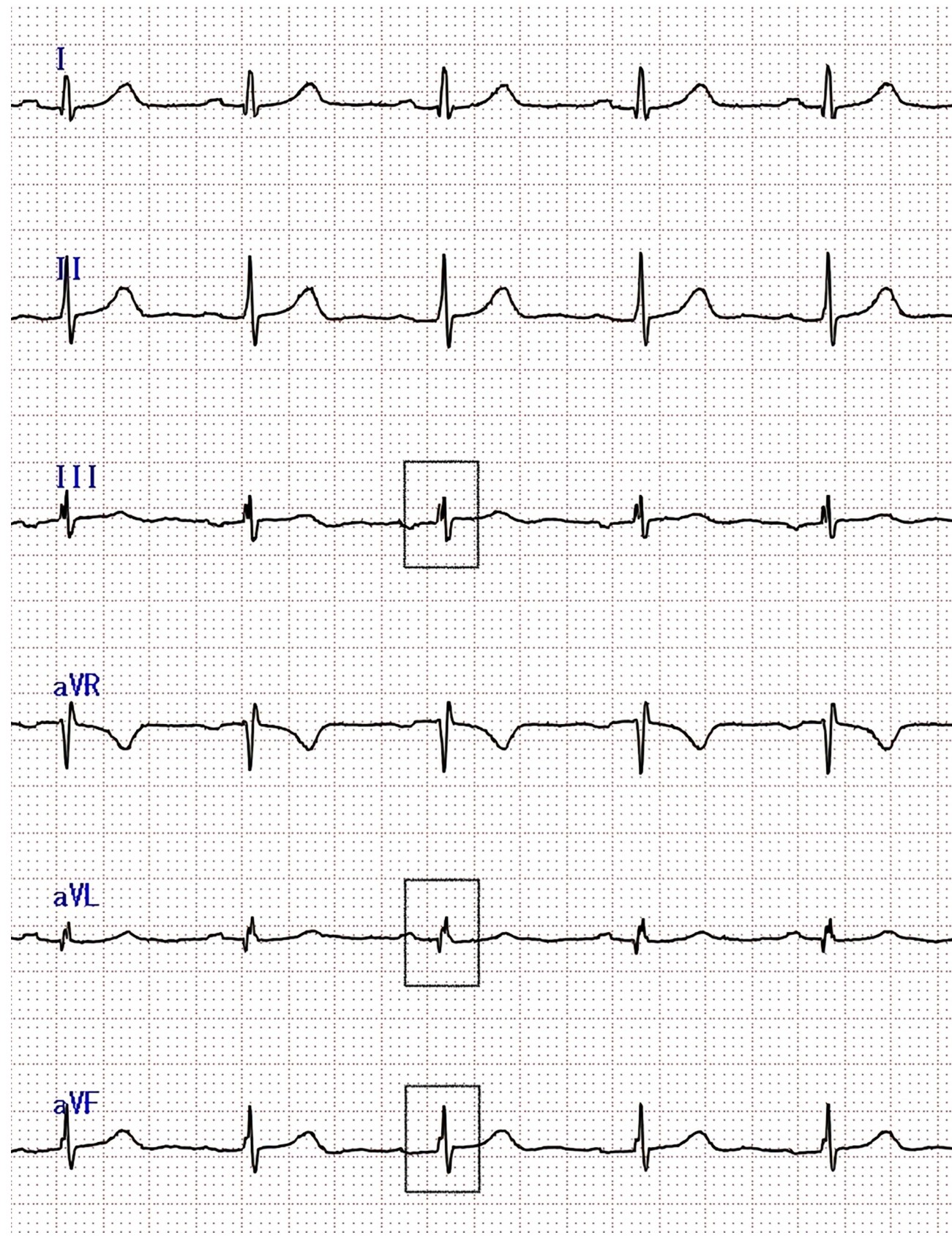

**Fig 1. Example of fQRS in a patient with SLE.** A notch in the R wave is presented in III, aVL, and aVF. The fQRS complex was defined by the presence of a notch in the R wave in 2 contiguous leads (III and aVF). (fQRS: fragmented QRS; SLE: systemic lupus erythematosus).

respectively) than in the fQRS(-) group. No significant differences were found between the fQRS(+) and fQRS(-) groups with respect to other clinical features.

During the main analysis, the regression coefficient of fQRS for SLEDAI-2K was 2.69 (95% confidence interval [CI], 0.76–4.61; p = 0.008) in reference to fQRS(-) (Table 2). During secondary analysis, the regression coefficient of fQRS for nephritis was 0.94 (95% CI, 0.28–1.74; p = 0.014) in reference to fQRS(-). There were no significant associations between fQRS and the blood tests or evidence of end-organ involvement other than nephritis (Tables 3 and 4). According to the first sensitivity analysis, based on the ECG results determined by each of the cardiologists (ET and JS), SLEDAI-2K was significantly higher for the fQRS(+) group than for the fQRS(-) group. According to the evaluation by ET, 27 patients (61%) had fQRS, and the regression coefficient of fQRS for SLEDAI-2K was 2.80 (95% CI, 0.90–4.70; p = 0.005) in

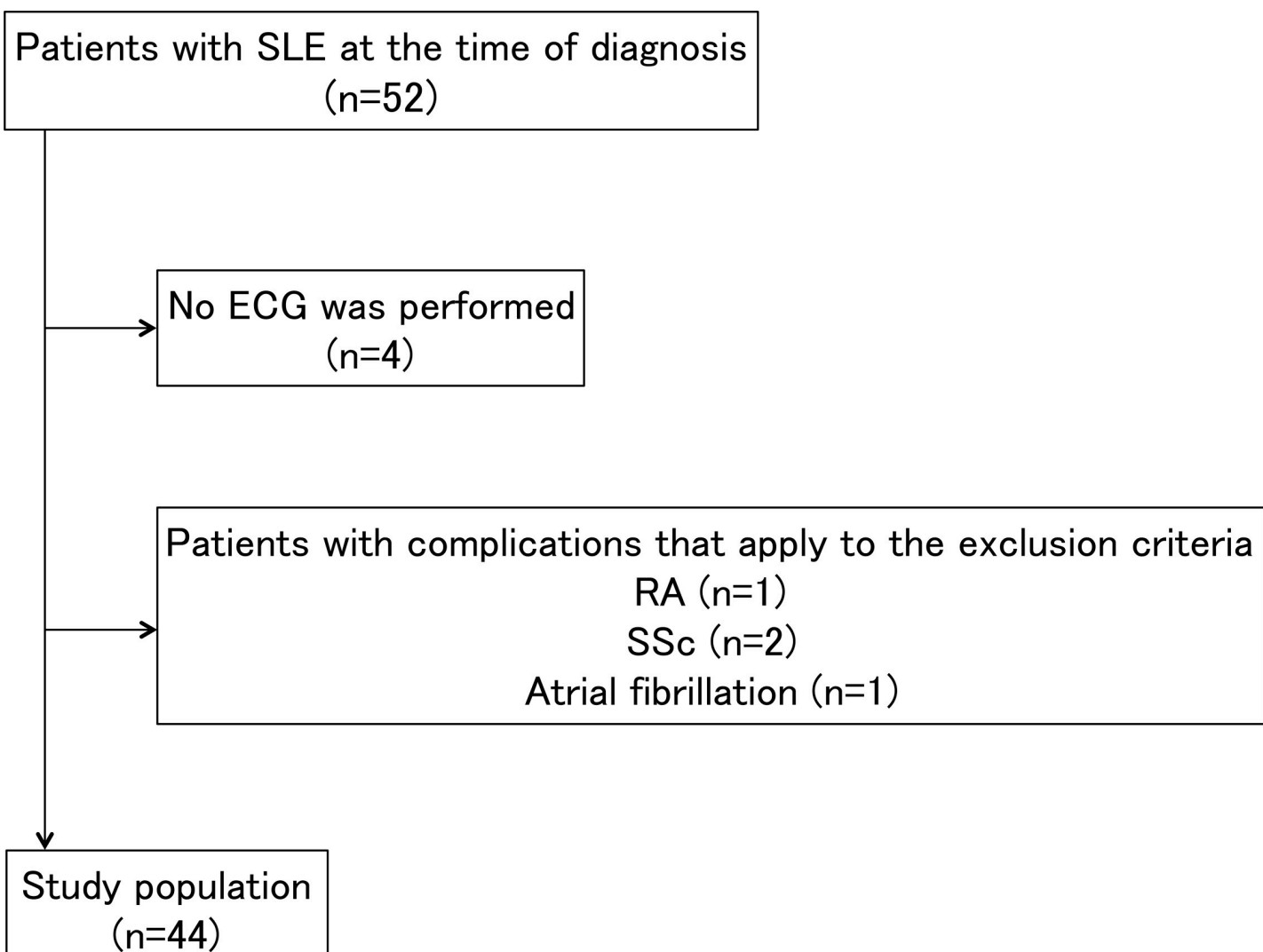

**Fig 2. Patient flow chart.** (SLE: systemic lupus erythematosus; ECG: electrocardiogram; RA: rheumatoid arthritis; SSc: systemic sclerosis).

**Table 1. Baseline characteristics of SLE patients.**

| | Total (n = 44) | fQRS(+) (n = 26) | fQRS(-) (n = 18) | Missing data | *p*-value |
|---|---|---|---|---|---|
| **Demographics** | | | | | |
| Age, mean (SD) | 39.5 (27.5) | 44.2 (22.9) | 41.2 (13.2) | 0 | 0.47 |
| Women, n (%) | 37 (84.1) | 19 (73.1) | 18 (100) | 0 | 0.031 |
| Time period from onset to diagnosis, median (IQR), months | 3.0 (2.0–14.8) | 3.0 (1.8–14.3) | 3.0 (2.0–23.5) | 0 | 0.52 |
| SBP, median (IQR), mmHg | 115 (102–126) | 115 (102–129) | 115 (102–121) | 1 | 0.73 |
| DBP, median (IQR), mmHg | 69 (60–76) | 70 (59–76) | 69 (60–86) | 1 | 0.97 |
| Smoking status, n (%) | 5 (11.6) | 4 (16.0) | 1 (5.6) | 1 | 0.38 |
| **Comorbid conditions** | | | | | |
| Hypertension, n (%) | 4 (9.09) | 4 (15.4) | 0 (0) | 0 | 0.133 |
| Dyslipidemia, n (%) | 2 (4.6) | 2 (7.7) | 0 (0) | 0 | 0.51 |
| Diabetes mellitus, n (%) | 0 (0) | 0 (0) | 0 (0) | 0 | — |
| **Laboratory measurements** | | | | | |
| Framingham Risk Score, median (IQR) | -2 (-7–3) | -3 (-7–3) | -1 (-7–4) | 4 | 0.72 |
| LDL cholesterol, median (IQR), mg/dL | 91 (80–111) | 93 (79–114) | 71 (80–107) | 2 | 0.78 |
| HDL cholesterol, median (IQR), mg/dL | 36 (28–50) | 34 (26–43) | 39 (29–54) | 3 | 0.33 |
| Triglyceride, median (IQR), mg/dL | 137 (103–201) | 145 (109–213) | 120 (93–181) | 3 | 0.47 |
| Uric acid, median (IQR), mg/dL | 4.9 (4.0–5.8) | 5.0 (4.0–6.2) | 4.3 (3.4–5.2) | 6 | 0.156 |
| HbA1c, median (IQR), % | 5.6 (5.3–6.1) | 5.6 (5.2–5.9) | 5.7 (5.3–6.1) | 11 | 0.61 |
| Anti-dsDNA antibody, median (IQR), EU/mL | 37.5 (5.5–236.2) | 65.9 (7.2–380.0) | 28.1 (4.8–169.7) | 1 | 0.34 |
| Anti-beta-2-GP antibody, n (%) | 5 (12.2) | 3 (12.5) | 2 (11.8) | 3 | 1.00 |
| Anti-SS-A antibody, n (%) | 30 (69.8) | 17 (68.0) | 13 (72.2) | 1 | 1.00 |
| Anti-U1-RNP antibody, n (%) | 12 (37.5) | 4 (22.2) | 8 (57.1) | 12 | 0.068 |
| CH50, median (IQR), U/mL | 13 (6–26) | 12 (6–22) | 22 (7–34) | 6 | 0.21 |
| C3, median (IQR), mg/dL | 55.7 (36.0–80.8) | 48.2 (34.2–90.8) | 59.5 (41.3–80.2) | 1 | 0.38 |
| C4, median (IQR), mg/dL | 9.1 (4.3–14.8) | 7.4 (3.8–14.6) | 11.7 (4.4–26.7) | 1 | 0.172 |
| IC-C1q, median (IQR), µg/mL | 4.0 (1.5–8.4) | 4.0 (1.5–9.0) | 3.4 (1.5–8.2) | 3 | 0.61 |
| CRP, median (IQR), mg/dL | 0.69 (0.16–1.64) | 1.15 (0.22–1.73) | 0.42 (0.09–1.83) | 0 | 0.37 |
| **Organ involvement** | | | | | |
| SLEDAI-2K, median (IQR) | 14 (10–20) | 18 (12–22) | 9 (8–15) | 0 | <0.001 |
| Cutaneous, n (%) | 28 (63.6) | 18 (69.2) | 10 (55.6) | 0 | 0.52 |
| Arthritis, n (%) | 29 (65.9) | 17 (65.4) | 12 (66.7) | 0 | 1.00 |
| Myositis, n (%) | 1 (2.3) | 1 (3.9) | 0 (0) | 0 | 1.00 |
| Pericarditis, n (%) | 4 (9.1) | 2 (7.7) | 2 (11.1) | 0 | 1.00 |
| Pleuritis, n (%) | 8 (18.2) | 6 (23.1) | 2 (11.1) | 0 | 0.44 |
| Lupus enteritis, n (%) | 4 (9.1) | 4 (15.4) | 0 (0) | 0 | 0.133 |
| Lupus cystitis, n (%) | 1 (2.3) | 1 (3.9) | 0 (0) | 0 | 1.00 |
| Vasculitis, n (%) | 1 (2.3) | 1 (3.9) | 0 (0) | 0 | 1.00 |
| Renal disorder, n (%) | 18 (40.9) | 14 (53.9) | 4 (22.2) | 0 | 0.061 |
| Neurologic disorder, n (%) | 5 (11.4) | 4 (15.4) | 1 (5.6) | 0 | 0.63 |
| Leukopenia, n (%) | 20 (45.5) | 12 (46.2) | 8 (44.4) | 0 | 1.00 |
| Thrombocytopenia, n (%) | 8 (18.2) | 6 (23.1) | 2 (11.1) | 0 | 0.44 |
| **Electrocardiogram data** | | | | | |
| HR, median (IQR), bpm | 79 (70–89) | 78 (70–89) | 82 (76–89) | 0 | 0.37 |
| QRS duration, median (IQR), ms | 83 (78–90) | 84 (78–90) | 82 (78–89) | 0 | 0.91 |
| QTc interval, median (IQR), ms | 412 (400–423) | 414 (399–425) | 409 (401–421) | 0 | 0.44 |

*(Continued)*

**Table 1.** (Continued)

| | Total (n = 44) | fQRS(+) (n = 26) | fQRS(-) (n = 18) | Missing data | *p*-value |
|---|---|---|---|---|---|
| **Echocardiographic data** | | | | | |
| LVEF, median (IQR), % | 66 (62–71) | 63 (62–70) | 66 (61–74) | 12 | 0.39 |
| LVDd, median (IQR), mm | 46 (43–49) | 47 (43–53) | 46 (43–46) | 11 | 0.100 |
| LVDs, median (IQR), mm | 29 (27–33) | 29 (27-–36) | 28 (25–30) | 11 | 0.164 |
| RVSP, median (IQR), mmHg | 25.1 (22.1–30.0) | 25.0 (22.0–29.8) | 25.2 (22.1–31.4) | 12 | 0.83 |

IQR: interquartile range; SD: standard deviation; fQRS: fragmented QRS; SBP: systolic blood pressure; DBP: diastolic blood pressure; LDL: low-density lipoprotein; HDL: high-density lipoprotein; HbA1C: glycated hemoglobin; Anti-SS-A: anti-Sjogren syndrome antibody A; SLEDAI-2K: Systemic Lupus Erythematosus Disease Activity Index 2000; beta-2-GP: β2-glycoprotein I antibody; CH50: hemolytic complement activity; C3/4: complement 3/4; IC-C1q: C1q-binding immune complexes; CRP: C-reactive protein; HR: heart rate; QTc interval: corrected QT interval; LVEF: left ventricular ejection fraction; LVDd: left ventricular end-diastolic dimension; LVDs: left ventricular end-systolic dimension; RVSP: right ventricular systolic pressure

reference to fQRS(-). According to the evaluation by JS, 27 patients (61%) had fQRS, and the regression coefficient of fQRS for SLEDAI-2K was 3.05 (95% CI, 1.13–4.96; p = 0.003) in reference to fQRS(+). Inter-observer variabilities showed great agreement between the 2 blinded experienced cardiologists reading fQRS, with a 90.9% consensus (κ = 0.81; 95% CI, 0.63–0.98; p<0.001). During the second sensitivity analysis, the ECGs were evaluated and analyzed at 100% magnification on paper, and the SLEDAI-2K was significantly higher for the fQRS(+) group. There was good agreement between the ECGs evaluated on paper and those evaluated on the monitor when magnified at 400%. Twenty-two patients (50%) had fQRS. The regression coefficient of fQRS for SLEDAI-2K was 2.86 (95% CI, 1.15–4.56; p = 0.002) in reference to fQRS(-). The consensus was 90.9% (κ = 0.82; 95% CI, 0.65–0.99; p<0.001). During the third sensitivity analysis, we excluded 4 patients with existing cardiovascular risks. The regression coefficient of fQRS for SLEDAI-2K was 4.52 (95% CI, 0.32–8.73; p = 0.036) in reference to fQRS(-).

## Discussion

This study investigated the relationship between disease activity in patients with SLE and fQRS at diagnosis after adjustment for age, sex, and time period from the estimated onset date to the date of diagnosis. The fQRS(+) rate among patients with SLE at diagnosis was 59% and 33% of the fQRS(+) patients disappeared fQRS after immunosuppressive therapy. The SLEDAI-2K at diagnosis was significantly higher for the fQRS(+) group than for the fQRS(-) group. Additionally, the fQRS(+) group showed a high rate of nephritis. This study is the first report on an association between ECG abnormalities and disease activity at the time of diagnosis in patients with SLE. Therefore, fQRS should be used to detect subclinical myocardial involvement in patients with SLE.

**Table 2. Association between fQRS and SLEDAI-2K in the multilinear regression analysis.**

| | Regression coefficient | SE | 95% CI | *p*-value |
|---|---|---|---|---|
| fQRS(-) | Reference | – | – | – |
| fQRS(+) | 2.69 | 0.95 | 0.76–4.61 | 0.008 |

SLEDAI-2K: Systemic Lupus Erythematosus Disease Activity Index 2000; fQRS: fragmented QRS; SE: standard error; CI: confidence interval

**Table 3. Association between fQRS and laboratory measurements in the multilinear regression analysis.**

|  | Regression coefficient | SE | 95% CI | *p*-value |
|---|---|---|---|---|
| Anti-dsDNA antibody (EU/mL) |  |  |  |  |
| fQRS(-) | Reference | – | – | – |
| fQRS(+) | 33.10 | 25.72 | -19.00–85.16 | 0.21 |
| CH50 (U/mL) |  |  |  |  |
| fQRS(-) | Reference | – | – | – |
| fQRS(+) | -4.54 | 2.69 | -10.01–0.93 | 0.10 |
| C3 (mg/dL) |  |  |  |  |
| fQRS(-) | Reference | – | – | – |
| fQRS(+) | -8.10 | 6.18 | -20.61–4.41 | 0.20 |
| C4 (mg/dL) |  |  |  |  |
| fQRS(-) | Reference | – | – | – |
| fQRS(+) | -3.96 | 2.15 | -8.32–0.41 | 0.07 |

fQRS: fragmented QRS; SE: standard error; CI: confidence interval; C3/4: complement 3/4; CH50: hemolytic complement activity

The fQRS can be caused by zigzag conduction around myocardial scarring resulting from a previous myocardial infarction or inflammation [12]. It has been reported that fQRS showed high sensitivity and a high negative predictive value for detecting myocardial scars in patients with CAD identified via myocardial single-photon emission tomography and CMR [11,15]. The fQRS of patients with sarcoidosis was also reported to be associated with cardiac involvement detected using late gadolinium enhancement on CMR to identify myocardial fibrosis [14]. A previous report indicated that myocardial edema, defined as an increased T2 ratio on CMR appearing as myocardial infarction and inflammation, was observed in 63% patients who had not undergone treatment for SLE at the time of diagnosis. This result is similar to ours, which indicates that 59% of untreated patients with SLE had fQRS at the time of diagnosis [24]. A previous study on fQRS in patients with SLE after treatment interventions indicated that the fQRS(+) rate was 41%, which was lower than that in our study [22]. Because 33% of the fQRS(+) patients disappeared fQRS after immunosuppressive therapy, the differences in results may be explained by the disappearance of fQRS with the initiation of therapeutic

**Table 4. Association between fQRS and organ involvement according to the logistic regression analysis.**

|  | N (%) | Absolute Risk Difference, % (95% CI) | Adjusted Odds Ratio (95% CI) | *p*-value |
|---|---|---|---|---|
| Cutaneous |  |  |  |  |
| fQRS(-) | 10/18 (55.6) | NA | 1 [Reference] | – |
| fQRS(+) | 18/26 (69.2) | 14.29 (-15.97–44.54) | 1.07 (0.27–4.32) | 0.92 |
| Arthritis |  |  |  |  |
| fQRS(-) | 12/18 (66.7) | NA | 1 [Reference] | – |
| fQRS(+) | 17/26 (65.4) | -1.48 (-31.97–29.21) | 1.08 (0.26–4.51) | 0.92 |
| Renal disorder |  |  |  |  |
| fQRS(-) | 4/18 (22.2) | NA | 1 [Reference] | – |
| fQRS(+) | 14/26 (53.9) | 31.62 (4.49–58.75) | 6.54 (1.47–29.05) | 0.014 |
| Leukopenia |  |  |  |  |
| fQRS(-) | 8/18 (44.4) | NA | 1 [Reference] | – |
| fQRS(+) | 12/26 (46.2) | 1.67 (-27.49–30.82) | 1.13 (0.30–4.32) | 0.85 |

fQRS: fragmented QRS; CI: confidence interval; NA: not applicable

interventions for SLE. Reversible fQRS has also been reported in several studies on myocardial infarctions [25,26]. Therefore, we hypothesized that fQRS would be expressed by not only irreversible myocardial replacement fibrosis because of myocardial infarction or inflammation but also by ischemia or inflammation themselves in patients with SLE in the absence of typical cardiac symptoms.

Our results showed that fQRS and SLE disease activity are related, suggesting that fQRS exists in patients with SLE disease activity high enough to cause myocardial injury. Cardiac involvement in SLE including myocarditis and infarction may be mediated by immunological mechanisms. Immunofluorescence studies indicated that fine granular immune complexes and complement deposits were detected in the walls and perivascular tissues of myocardial blood vessels [27]. Inflammatory cell infiltration in the coronary artery has also been observed in the autopsy results of patients with SLE who died secondary to myocardial infarction [1]. These pathologic findings suggested that lupus myocarditis and infarction are primarily caused by circulating immune complexes that mediate inflammation. Moreover, increasing T2 ratios indicated that myocardial inflammation and infarction detected on CMR were significantly more frequent in patients with SLE who had a high disease activity [8,9]. Furthermore, in the LUMINA study, which included a multi-ethnic United States cohort, it was reported that a high disease activity at the time of diagnosis was related to the development of myocarditis [5]. A cohort of individuals with lupus in Toronto was found to have higher baseline and recent disease activity scores that increased with the occurrence of CAD-related events [6]. Therefore, we believe that higher SLE disease activity results in increased myocardial involvement; therefore, fQRS is relevant to SLEDAI-2K, which can comprehensively evaluate systemic organ damage mediated by immunological mechanisms.

The association between fQRS and lupus nephritis in this study may be explained by an immunological mechanism related to *in situ* immune complexes (ICs). The cause of lupus nephritis involves not only circulating ICs but also *in situ* ICs, such as the binding of anti-dsDNA antibody to α-actinin [28,29]. It has been speculated that α-actinin is also present in cardiac muscle and that myocardial damage occurs via a similar mechanism. Although α-actinin is frequently present in skeletal muscle, it was not associated with fQRS and myositis in this study [30]. Since myositis usually presents with atypical and non-specific symptoms such as fatigue, it was not detected in this study and was possibly unrecognized as being related to fQRS [31]. Recent epidemiological studies showed that SLE patients with lupus nephritis have a significantly higher risk of MI and CVD mortality than those without lupus nephritis and that they had carotid atherosclerotic plaques twice as often as non-nephritis SLE patients and population controls [32,33]. Therefore, lupus nephritis may be related to fQRS because of both the kidney and the heart being targets of *in situ* ICs.

There are 3 main strengths of this study. First, to the best of our knowledge, this is the first study to use a multivariate analysis adjusted for appropriate confounding factors to evaluate fQRS in SLE. In a previous study using univariate analysis, an association was found between fQRS and disease duration [22]. Therefore, we defined the period from the onset of symptoms to the diagnosis as the confounding factor. Second, it was revealed that fQRS disappears after immunosuppressive therapy, suggesting that the mechanism of fQRS by SLE involves reversible inflammation and ischemia. This indicates that fQRS is a marker that not only identifies myocardial involvement at the time of diagnosis but also can evaluate changes due to treatment intervention, and leverage the strength of ECG that can be repeatedly evaluated non-invasively. Third, 3 sensitivity analyses were performed to determine the main outcome, and all results were compatible with each other. There was a significant difference in the results as noted by the 2 blinded cardiologists. Additionally, excellent inter-rater agreement was achieved by each of the cardiologists. A significant relationship was reproduced when fQRS was printed at 100%

magnification and read on paper. This was shown to be useful in environments where the ECG could not be read on a monitor. Even after excluding 4 patients with classical cardiovascular risk factors, a significant association between fQRS and SLEDAI-2K was maintained.

There are also 3 main limitations to this study. First, 4 patients were excluded because they had no ECG measurements available at the time of diagnosis. However, despite the small sample size in our study, there was no bias regarding the main outcome of SLEDAI-2K. Second, this was a retrospective evaluation of a series of medical records. However, data deficiencies for the main outcome and the confounders were not admitted. Third, coexistence of myocardial involvement not contributed by SLE at the time of diagnosis cannot be denied. However, there was no significant difference between the fQRS(+) and fQRS(-) groups with respect to the Framingham Risk Score and coronary risk factors (hypertension, diabetes, dyslipidemia, LDL cholesterol, HDL cholesterol, triglyceride, uric acid level, and HbA1c). Furthermore, after excluding patients with classical cardiovascular risk factors, the sensitivity analysis showed a significant association between fQRS and SLEDAI-2K.

There are 2 clinical implications of the study findings. First, fQRS defined by ECG is suitable for screening for myocardial involvement in patients with SLE because it can be measured immediately and in any environment. As most myocardial involvement in patients with SLE is subclinical, it is necessary to evaluate all patients with SLE. Although CMR is highly sensitive for detecting subclinical cardiac involvement in patients with SLE, its use is restricted by its high medical costs, medical infrastructure, and patient condition [7]. Therefore, it is impossible to perform CMR at the time of diagnosis for all patients with SLE. The mechanisms of examination are different for fQRS, which evaluates conduction disturbances electrophysiologically, and CMR, which provides qualitative evaluations of the myocardial tissue. However, there was a good correlation between fQRS and late gadolinium enhancement for various cardiomyopathies identified using CMR[14,34,35]. Therefore, even for patients with SLE, fQRS is useful for routine evaluations of myocardial injury. A mechanized learning approach to detect fQRS has been attempted and is expected to provide more objective and simple indications [36]. The fQRS can be used as a suitable parameter for routinely screening myocardial involvement in patients with SLE. Second, fQRS could be a predictor of long-term cardiac function and arrhythmia in patients with SLE. It represents myocardial replacement fibrosis, which appears at the sites of prior inflammation or infarction and can be associated with ventricular dysfunction and the development of congestive heart failure. Additionally, myocardial scars detected using fQRS are a substrate for re-entrant ventricular tachyarrhythmia [37]. A meta-analysis of patients with reduced EF showed that fQRS was associated with increased all-cause mortality up to 1.63-fold, as well as increased major arrhythmic events up to 1.74-fold [38]. Another meta-analysis of patients with acute coronary syndrome reported that fQRS was an independent predictor of mortality, major adverse cardiovascular events, and deteriorating LV function [15]. In the future, we plan to perform a longitudinal study to clarify whether fQRS can predict cardiovascular disease in patients with SLE.

## Conclusion

Our results demonstrated that fQRS(+) is associated with high disease activity in patients with SLE. We believe that fQRS can detect subclinical myocardial involvement in patients with SLE and could be a predictor of long-term cardiac function and arrhythmia.

## Supporting information

**S1 Fig. Variations of fragmented QRS.**
(TIFF)

## Acknowledgments

We are grateful to Takeo Isozaki, Kuninobu Wakabayashi, Sakiko Isojima, Hidekazu Furuya, Takahiro Tokunaga, Mayu Saito, Masayu Umemura, Nao Oguro, Yoko Miura, Sho Ishii, Shinya Seki, Shinichiro Nishimi, Airi Nishimi, Yuzo Ikari, Tomoki Hayashi, Mika Hatano, and Akihiro Nakamura for their cooperation with data collection. We are indebted to Noboru Murata (Kikuna Memorial Hospital) for critically reviewing the manuscript.

## Author Contributions

**Conceptualization:** Masahiro Hosonuma.

**Data curation:** Ryo Yanai.

**Formal analysis:** Masahiro Hosonuma.

**Investigation:** Eiji Toyosaki, Jumpei Saito.

**Methodology:** Nobuyuki Yajima.

**Project administration:** Masahiro Hosonuma, Nobuyuki Yajima.

**Supervision:** Nobuyuki Yajima, Ryo Takahashi, Ryo Yanai, Taka-aki Matsuyama, Kengo Kusano, Hiroshi Morita.

**Writing – original draft:** Masahiro Hosonuma.

**Writing – review & editing:** Nobuyuki Yajima, Taka-aki Matsuyama, Kengo Kusano, Hiroshi Morita.

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
