## [Decision Letter · Decision Letter 0]

15 Nov 2019

PONE-D-19-29766

Fragmented QRS complex in patients with systemic lupus erythematosus at the time of diagnosis and its relationship with disease activity

PLOS ONE

Dear Dr. Yajima,

Thank you for submitting your manuscript to PLOS ONE. After careful consideration, we feel that it has merit but does not fully meet PLOS ONE’s publication criteria as it currently stands. Therefore, we invite you to submit a revised version of the manuscript that addresses the points raised during the review process.

The topic of this article is of potential interest, but reviewers pointed out several important insights. Especially, mechanistic relationship between cardiac fibrosis and disease activity needs to be discussed.

We would appreciate receiving your revised manuscript by Dec 30 2019 11:59PM. To enhance the reproducibility of your results, we recommend that if applicable you deposit your laboratory protocols in protocols.io, where a protocol can be assigned its own identifier (DOI) such that it can be cited independently in the future. For instructions see: http://journals.plos.org/plosone/s/submission-guidelines#loc-laboratory-protocols

We look forward to receiving your revised manuscript.

Kind regards,

Masataka Kuwana, MD, PhD

Academic Editor

PLOS ONE

Journal Requirements:

2. In the ethics statement in the manuscript and in the online submission form, please provide additional information about the patient records used in your retrospective study, including: a) the date range (month and year) during which patients' medical records were accessed; b) the date range (month and year) during which patients whose medical records were selected for this study sought treatment; and c) the source of the medical records analyzed in this work (e.g. hospital, institution or medical center name).

3. Your ethics statement must appear in the Methods section of your manuscript. If your ethics statement is written in any section besides the Methods, please move it to the Methods section and delete it from any other section. Please also ensure that your ethics statement is included in your manuscript, as the ethics section of your online submission will not be published alongside your manuscript.

I have read the journal's policy and the authors of this manuscript have the following competing interests: [Hiroshi Morita is affiliated with the endowment department of Japan Medtronic, Inc. The other authors declare that there is no conflict of interest.]

Reviewers' comments:

Reviewer's Responses to Questions

**Comments to the Author**

1. Is the manuscript technically sound, and do the data support the conclusions?

Reviewer #1: Yes

Reviewer #2: Partly

2. Has the statistical analysis been performed appropriately and rigorously? 

Reviewer #1: Yes

Reviewer #2: N/A

3. Have the authors made all data underlying the findings in their manuscript fully available?

Reviewer #1: Yes

Reviewer #2: No

4. Is the manuscript presented in an intelligible fashion and written in standard English?

Reviewer #1: Yes

Reviewer #2: Yes

5. Review Comments to the Author

Reviewer #1: Hosomuma M et al evaluated relationship between disease activity in SLE patients and fQRS at their diagnosis. Overall the manuscript is on important topic and is written and analyzed with clarity and conviction. However, I have some concerns which authors need to answer.

Major

1. Authors concluded that a significant association between fQRS presence and disease activity by Table 1 (page 26). Although I agree with author's conclusion by 2.69 of regression coefficient and the low p-value, 95% CI was started with less than 1.0. I wonder their conclusion is always confident.

2. As authors mentioned in Introduction section, CVD leads to poor prognosis and early recognition by non-invasive method is highly valuable in clinical settings. By this point of view, this manuscript gives important message. However, my concern is fQRS is really associated with cardiac ischemia or inflammation, systemic disease activity, and future development of CVD. Please show some additional data if authors have conducted further analysis by using Cardiac MRI for patients with or without fQRS, serial change of fQRS after immunosuppressive treatment (Dose it disappear?), and cumulative rate of CVD depending on fQRS presence.

Reviewer #2: There are studies on this subject. Therefore, the study does not reveal any new information. SLE is a chronic disease and FQRS is associated with interstitial fibrosis. Therefore, I do not think FQRS is associated with disease activation.

6. PLOS authors have the option to publish the peer review history of their article (what does this mean?). If published, this will include your full peer review and any attached files.

Reviewer #1: No

Reviewer #2: No

---

## [Author Response · Author response to Decision Letter 0]

5 Dec 2019

Responses to editors’ and reviewers’ comments on PONE-D-19-29766

We are grateful to Editorial Board Member and two reviewers for their comments and invaluable suggestions. We have incorporated new data after performing additional analysis, and carefully revised the manuscript accordingly. We believe that the revised manuscript, with the changes and new data added, has taken into consideration essentially all of the comments, and hope that will have been improved to the satisfaction of the editors and reviewers. Please find below our point-by-point response to each of the comments made by the reviewers. 

Reviewer comments:

Reviewer #1: Hosomuma M et al evaluated relationship between disease activity in SLE patients and fQRS at their diagnosis. Overall the manuscript is on important topic and is written and analyzed with clarity and conviction. However, I have some concerns which authors need to answer.

Major

1. Authors concluded that a significant association between fQRS presence and disease activity by Table 1 (page 26). Although I agree with author's conclusion by 2.69 of regression coefficient and the low p-value, 95% CI was started with less than 1.0. I wonder their conclusion is always confident.

We appreciate the opportunity to clarify this point. In case of a multiple linear regression analysis, if the 95% CI contains zero, and the effect will not be significant. On the other hand, In case of a Logistic regression analysis, if the 95% CI contains 1.0, and the effect will not be significant (BMJ 2013;347:f4373). During the main analysis, a multiple linear regression analysis was conducted to assess the association between fQRS and SLE activity, and the 95% CI was not contains zero (0.76–4.61). Therefore, we concluded that it was statistically significant.

2. As authors mentioned in Introduction section, CVD leads to poor prognosis and early recognition by non-invasive method is highly valuable in clinical settings. By this point of view, this manuscript gives important message. However, my concern is fQRS is really associated with cardiac ischemia or inflammation, systemic disease activity, and future development of CVD. Please show some additional data if authors have conducted further analysis by using Cardiac MRI for patients with or without fQRS, serial change of fQRS after immunosuppressive treatment (Dose it disappear?), and cumulative rate of CVD depending on fQRS presence.

We appreciate your suggestions. Based on the reviewer’s helpful suggestions, in order to clarify whether it was serial change of fQRS after immunosuppressive treatment, we investigated whether ECG was performed after treatment. Of the 26 patients who were fQRS positive at the time of diagnosis, 18 patients were followed, and 6 patients (33%) disappeared fQRS. This suggests that fQRS was caused by not only irreversible fibrosis but also inflammation and responded to immunosuppressive therapy. Therefore, we think FQRS is associated with acute disease activity. We have included these additional methods, results and related discussion in the revised version of the manuscript (page 7, lines16-17; page 9, lines 14-16; page 11, lines 3-5; page 11, lines 22-page 12, line 3; page 13, lines 15-20).

In the future study, it is necessary to increase the number of cases and clarify what causes the difference between cases where fQRS disappears and cases where it does not. In addition, it has been reported that myocardial inflammation detected by cardiac MRI disappears after treatment (Hinojar R et al. Int J Cardiol. 2016 Nov 1;222:717-726). Unfortunately, we did not measure CMR at the time of diagnosis. In the Limitations section, we have acknowledged this issue. 

Reviewer #2: 

There are studies on this subject. Therefore, the study does not reveal any new information. 

We appreciate the opportunity to clarify this point. 

The novelties of our methods are 

#1 using a multivariate analysis adjusted for appropriate confounding factors.

#2 the subject is an SLE at the time of diagnosis that is not affected by treatment.

#3 evaluating the relationship between disease activity and fQRS.

#4 longitudinal assessment of changes in fQRS after treatment.

The novelties of our results are 

#1 the first report on an association between ECG abnormalities and disease 

　　activity in patients with SLE.

#2 the incidence of fQRS (+) patients who disappeared after immunosuppressive 

　　therapy.

SLE is a chronic disease and FQRS is associated with interstitial fibrosis. Therefore, I do not think FQRS is associated with disease activation.

Please see our responses to Reviewer #1 Comments 2 described above. Because there are cases where fQRS disappears after treatment, fQRS was caused by not only irreversible fibrosis but also inflammation. Therefore, we think FQRS is associated with acute disease activity.

---

## [Decision Letter · Decision Letter 1]

12 Dec 2019

Fragmented QRS complex in patients with systemic lupus erythematosus at the time of diagnosis and its relationship with disease activity

PONE-D-19-29766R1

Dear Dr. Yajima,

We are pleased to inform you that your manuscript has been judged scientifically suitable for publication and will be formally accepted for publication once it complies with all outstanding technical requirements.

With kind regards,

Masataka Kuwana, MD, PhD

Academic Editor

PLOS ONE

Additional Editor Comments (optional):

Reviewers' comments:

Reviewer's Responses to Questions

**Comments to the Author**

1. If the authors have adequately addressed your comments raised in a previous round of review and you feel that this manuscript is now acceptable for publication, you may indicate that here to bypass the “Comments to the Author” section, enter your conflict of interest statement in the “Confidential to Editor” section, and submit your "Accept" recommendation.

Reviewer #1: All comments have been addressed

2. Is the manuscript technically sound, and do the data support the conclusions?

Reviewer #1: Yes

3. Has the statistical analysis been performed appropriately and rigorously? 

Reviewer #1: Yes

4. Have the authors made all data underlying the findings in their manuscript fully available?

Reviewer #1: Yes

5. Is the manuscript presented in an intelligible fashion and written in standard English?

Reviewer #1: Yes

6. Review Comments to the Author

Reviewer #1: (No Response)

7. PLOS authors have the option to publish the peer review history of their article (what does this mean?). If published, this will include your full peer review and any attached files.

Reviewer #1: No

---

## [Editor Report · Acceptance letter]

17 Dec 2019

PONE-D-19-29766R1 

Fragmented QRS complex in patients with systemic lupus erythematosus at the time of diagnosis and its relationship with disease activity 

Dear Dr. Yajima:

I am pleased to inform you that your manuscript has been deemed suitable for publication in PLOS ONE. Congratulations! Your manuscript is now with our production department. 

With kind regards,

on behalf of

Prof. Masataka Kuwana 

Academic Editor

PLOS ONE